# COVID-19 Associated Invasive Pulmonary Aspergillosis: Diagnostic and Therapeutic Challenges

**DOI:** 10.3390/jof6030115

**Published:** 2020-07-22

**Authors:** Aia Mohamed, Thomas R. Rogers, Alida Fe Talento

**Affiliations:** 1Department of Microbiology, Our Lady of Lourdes Hospital Drogheda, A92 VW28 Co. Louth, Ireland; aiamohamed1987@gmail.com; 2Department of Clinical Microbiology, Trinity College Dublin, St. James’s Hospital Campus, D08 NHY1 Dublin, Ireland; rogerstr@tcd.ie; 3Department of Microbiology, Royal College of Surgeons, Ireland, D02 YN77 Dublin, Ireland

**Keywords:** COVID-19 pneumonia, invasive pulmonary aspergillosis, diagnosis, multi-triazole resistance, COVID-19 associated invasive pulmonary aspergillosis

## Abstract

*Aspergillus* co-infection in patients with severe coronavirus disease 2019 (COVID-19) pneumonia, leading to acute respiratory distress syndrome, has recently been reported. To date, 38 cases have been reported, with other cases most likely undiagnosed mainly due to a lack of clinical awareness and diagnostic screening. Importantly, there is currently no agreed case definition of COVID-19 associated invasive pulmonary aspergillosis (CAPA) that could aid in the early detection of this co-infection. Additionally, with the global emergence of triazole resistance, we emphasize the importance of antifungal susceptibility testing in order to ensure appropriate antifungal therapy. Herein is a review of 38 published CAPA cases, which highlights the diagnostic and therapeutic challenges posed by this novel fungal co-infection.

## 1. Introduction

Coronavirus disease 2019 (COVID-19), caused by severe acute respiratory syndrome coronavirus 2 (SARS-CoV-2), is a new viral respiratory infection first reported in Wuhan (Hubei province), China, at the end of 2019 [1]. Since then, more than 10 million confirmed COVID-19 cases, including more than half a million deaths, have been reported [2]. Although infection can vary from asymptomatic to mild upper respiratory infection, it can also lead to a severe pneumonia with acute respiratory distress syndrome (ARDS), requiring critical care and mechanical ventilation [3]. The case fatality rate varies by location and changes over time, and has been reported to be 0.2% in Germany and 7.7% in Italy, with elderly patients noted to have a greater risk of dying [4]. Recently, it was reported that 26% of patients admitted with severe COVID-19 infection died in intensive care [5].

SARS-CoV-2 infection leads to both innate and adaptive immune responses, which include a local immune response, recruiting macrophages and monocytes that respond to the infection, release cytokines, and prime adaptive T and B cell immune responses. In most cases, this process is capable of resolving the infection. However, in some cases, which present as severe COVID-19 infections, a dysfunctional immune response occurs, which can cause significant lung and even systemic pathology [6]. The diffuse alveolar lung damage and dysregulated immune response in severe COVID-19 pneumonia makes these patients vulnerable to secondary infections [6,7]. Viral, bacterial, and fungal co-infections have been reported in COVID-19 patients, and the early diagnosis of these co-infections is important in order to allow for the institution of appropriate antimicrobial therapy [8,9,10].

COVID-19 associated invasive pulmonary aspergillosis (CAPA) is a recently described syndrome that affects COVID-19 patients with ARDS who require critical care admission. With the global spread of COVID-19, as of 30 June 2020, 38 cases of CAPA have been reported. [11,12,13,14,15,16,17,18,19,20,21,22,23,24]. Here, we review these cases of CAPA so as to highlight the diagnostic and therapeutic challenges posed by this novel fungal co-infection.

## 2. Coronavirus and Aspergillosis

Coronaviruses are a large group of RNA viruses that infect humans, birds, bats, snakes, mice, and other animals. Seven known human coronaviruses (HCoVs) have been identified with 229E, OC43, NL63, and HKU1 more commonly detected. The first two account for approximately 15–29% of viral respiratory pathogens, with a relatively low virulence in humans [25,26]. The three other strains of HCoVs, namely severe acute respiratory syndrome coronavirus (SARS-CoV), Middle East respiratory syndrome coronavirus (MERS-CoV), and severe acute respiratory syndrome coronavirus 2 (SARS-CoV-2), have a different pathogenic potential, and have been shown to lead to higher mortality rates in humans [26,27].

To date, *Aspergillus* co-infection in patients with coronavirus infections is likely to have been under-diagnosed and under-reported, most likely due to lack of clinical awareness and diagnostic screening [28]. The published literature following severe acute respiratory syndrome (SARS) caused by SARS-CoV-1 has revealed only four cases of invasive aspergillosis (IA), all of which were diagnosed at post-mortem [29,30,31]. None of the four patients had a previous history of underlying immunocompromise, but they had received corticosteroids, which formed part of the treatment of patients with SARS in 2003. One of these patients was an intensive care physician who received several courses of methylprednisolone. The post-mortem findings in this patient were consistent with disseminated invasive aspergillosis with abscesses in multiple organs [29]. With regards to MERS-CoV, another HCoV that also causes severe respiratory infections, secondary bacterial infections have been reported [32], but a literature search failed to reveal published evidence of *Aspergillus* co-infection. This is most likely explained by the paucity of post-mortems performed on these patients, which were generally not done either for religious and cultural reasons, or to prevent environmental contamination with the subsequent infection of health-care workers [27].

Early reports from China documented *Aspergillus* spp. being isolated from the respiratory samples of patients with COVID-19 pneumonia, however there was no information on its clinical significance, or on the outcome of treatment of these patients [33,34]. Lescure et al. published a case series that detailed the first five imported cases of COVID-19 in France, whereby one of these five patients had severe COVID-19 pneumonia requiring critical care admission, and who was treated with triazoles when *Aspergillus flavus* was isolated from a tracheal aspirate [14].

As of 30 June 2020, 38 cases of CAPA have been reported from several countries, mostly in Europe, but the true incidence of this novel co-infection is unknown. All of the affected patients had been admitted to critical care because of COVID-19 pneumonia and ARDS, requiring ventilatory support. Thirty were males with a mean age of 65.9 (range 38–86, median 70). Table 1 summarizes these 38 cases, their pre-existing co-morbidities, their categorization using published definitions of IA, and their treatment and outcome.

## 3. Diagnosis of CAPA

The diagnosis of proven IA requires culture or histopathologic findings from biopsy or sterile site samples [37,38]. There are only six proven cases from the 38 reviewed here. One patient was suspected to have CAPA pre-mortem, when *A. fumigatus* was isolated from bronchoalveolar lavage fluid (BALF) and the serum galactomannan (GM) optical density index (ODI) was 8.6. Despite antifungal therapy (AFT), the patient succumbed to the infection, and the diagnosis of CAPA was confirmed at post-mortem. Another patient was diagnosed at post-mortem with histopathologic findings of fungal hyphae and spores in the lung tissue, further confirmed by nucleotide sequencing and identified as *A. penicillioides*. A stored peripheral blood sample revealed a GM ODI of 4.290 [15]. The other four proven cases were diagnosed by histopathological examination of the biopsy material taken from a bronchoscopy of the suspicious tracheobronchial lesions [13]. However, most patients with severe COVID-19 pneumonia are usually critically ill and hemodynamically unstable, which will preclude performing invasive procedures, such as bronchoscopy with a lavage or a lung biopsy. Furthermore, bronchoscopy is not recommended in patients with COVID-19 because of the risks this aerosol generating procedure imposes on both the patient and the attending healthcare worker, unless deemed life-saving [39]. According to current guidelines that are specific to different patient populations, the diagnosis of probable or putative invasive aspergillosis (IA) is made using a composite of host factors, clinical features, and mycological evidence of aspergillus infection [36,37,38]. Most patients with CAPA, including those with proven IA, did not have the host factors described for IA by the European Organization for Research and Treatment of Cancer and the Mycoses Study Group Education and Research Consortium (EORTC/MSGERC). Severe viral pneumonia is not considered a risk factor for invasive pulmonary aspergillosis, even though the structural damage as well as the dysregulated immune response can predispose to secondary co-infection with *Aspergillus* sp. [6,7]. An alternative diagnostic approach is to apply the clinical algorithm, which has been validated for the diagnosis of IA in patients in critical care [38], with severe COVID-19 infection, and the isolation of *Aspergillus* sp. from a BALF as the entry criteria. Recently, a panel of experts proposed case definitions for influenza-associated pulmonary aspergillosis that might also be considered for the classification of CAPA patients, while awaiting further histopathological studies that will provide more insight into the interaction between *Aspergillus* and SARS-CoV-2-infected lungs [35]. Patients with confirmed severe COVID-19 infection and pulmonary infiltrates on chest imaging should trigger investigation for the presence of *Aspergillus* infection by the culture of respiratory samples and/or the detection of GM either in serum or BALF, if and when bronchoscopy is performed. However, serum GM has a low sensitivity in non-neutropenic patients [40], and bronchoscopy may not be feasible. Endotracheal aspirates (ETA) are a potentially safer alternative investigative option, as their collection does not involve an aerosol generating procedure; however, their use for GM detection has not been validated. In previous reports, culturing *Aspergillus* spp. from ETA samples has been interpreted as colonization only, however when considered in conjunction with the clinical presentation and biomarkers, such as serum GM, this may suggest IA [41,42]. Of the 38 reported CAPA cases reviewed here, 16 and 14 patients had an *Aspergillus* sp. isolated from BALF and ETA samples, respectively, and another patient had *Aspergillus* sp. cultured from a sputum sample, with *A. fumigatus* being the most common species identified. The BALF/ETA GM indices were ≥1 in 16 of 23 patients, and the serum GM ODI was ≥0.5 in only 9 of 33 cases. New point-of-care tests for the detection of the *Aspergillus*-specific antigen or for GM from serum or BALF may also be useful as early evidence of CAPA in critically ill COVID-19 patients [43,44,45,46,47]. One patient with CAPA was reported to have the *Aspergillus* specific antigen detected from an ETA utilizing a lateral flow device [20]. The diagnostic performance of this lateral flow assay in the early diagnosis of IA in patients with severe influenza and/or COVID-19 is currently being investigated (ISRCTN51287266) [48]. Serum 1-3 β-d-glucan (BDG), a panfungal marker, was positive in 6 of 14 CAPA cases where BDG was reported. Although non-specific for *Aspergillus* infection, this biomarker is included as an indirect mycological criterion in the EORTC/MSGERC definitions; therefore, a positive BDG may help to support the diagnosis of CAPA [37] with an improved diagnostic performance when there are ≥2 positive results [49]. The detection of *Aspergillus* DNA using real-time PCR is another modality that may support the diagnosis of probable IA [37]. *Aspergillus* DNA was detected in 13 of 19 CAPA patients where real-time quantitative PCR was performed on either respiratory or serum samples.

The typical “halo sign” associated with IPA in neutropenic patients is uncommonly seen in non-neutropenic patients with IPA, where radiological imaging may show varying patterns from multiple pulmonary nodules to various non-specific findings, which include consolidation, cavitation, pleural effusions, ground glass opacities, tree-in-bud opacities, and atelectasis [37,50]. High resolution computed tomography (CT) is preferred to other imaging, such as chest radiographs [37,50]. Of the 38 reported CAPA cases, CT was performed in 15, where one patient was noted to have a reverse halo sign [20], six patients had ground glass opacities and varying sizes and numbers of nodules noted [11], while the others had findings “typical” of COVID-19 pneumonia. Patients with severe COVID-19 pneumonia in critical care are often clinically unfit for additional imaging, adding to the difficulty in interpreting the significance of the isolation of an *Aspergillus* sp. from upper respiratory tract samples.

Excluding the six proven CAPA cases, 18 of the remaining 32 cases reviewed here fulfilled the case definition of probable CAPA, as suggested by the expert panel [35]; 11 had putative IPA utilizing the *Asp*ICU criteria [38]; and one had probable CAPA utilizing EORTC/MSGERC definitions [37]. Two patients with chronic obstructive pulmonary disease (COPD) could be classified as probable CAPA, following definitions by Bulpa et al. for COPD patients [36]. We emphasize that definitions published by the EORTC/MSGERC are recommended only for research purposes, and should not be used for clinical decision making [37]. Perhaps a more pragmatic approach to the diagnosis of CAPA would be, in the setting of a patient with severe COVID-19 pneumonia in critical care, to combine ≥2 mycological criteria to include the following:
GM detection from serum/BALF/ETAIsolation of *Aspergillus* sp. from BALF/ETA/sputaSerum BDG detectionDetection of *Aspergillus* DNA by real time PCR in blood or respiratory samples

This approach may aid in the early institution of antifungal therapy.

## 4. Antifungal Treatment Strategies for CAPA

The clinical suspicion, or proven diagnosis, of *Aspergillus* co-infection should trigger the initiation of empiric or targeted antifungal therapy, respectively, even though its efficacy is not established. We note that only 13 of the 38 reported cases survived their infection, and those dying succumbed to multi-organ failure and sepsis. International treatment guidelines recommend the triazoles voriconazole or isavuconazole as the first-line treatment of IA [50,51]. The emergence of multi-triazole resistance in *A. fumigatus* challenges the efficacy of triazoles in the successful treatment of IPA [52,53,54], particularly in areas of high prevalence, and their use in such cases is associated with increased mortality [55]. Triazole resistance in *A. fumigatus* is causally linked to the use of triazole compounds that are structurally similar to those used in medical practice, as agricultural fungicides, or less commonly to prolonged triazole use in individual patients [54]. The former mechanism of resistance typically affects azole näive patients, and is characterized by elevated minimum inhibitory concentrations (MIC) of itraconazole, voriconazole, posaconazole, and isavuconazole. This underlines the importance of antifungal susceptibility testing (AFST) either through phenotypic or genotypic methods to detect triazole resistance, which will help direct the choice of treatment. Although cultures are generally known to have a poor diagnostic sensitivity [56], the ability to culture *Aspergillus* sp. will allow for the determination of MICs for triazoles. Recently, a four-well triazole resistance screening plate was validated for *A. fumigatus,* which can be useful in laboratories that do not have the capacity to perform the recommended broth microdilution methods for AFST [57,58,59]. Genotypic testing that utilizes molecular assays has also been evaluated to detect *Aspergillus* spp. and the common mutations associated with triazole resistance directly from clinical samples [60,61,62,63], which will allow for the rapid detection of a marker of resistance and guide treatment options. Twenty-two of the 38 CAPA cases reviewed here received a triazole-based AFT regimen either alone or in combination with an echinocandin or liposomal amphotericin B. Only seven cases reported susceptibility results based on either phenotypic testing and/or the detection of common mutations associated with triazole resistance using molecular techniques. Three cases were reported to be caused by a triazole-resistant *A. fumigatus,* all of which were confirmed to have the *cyp51A* TR_34_ L98H mutation [16,21,23]. Knowledge of the local epidemiology of triazole resistance is important to help guide the choice of therapy while awaiting susceptibility results. It has been recommended that for areas with triazole resistance rates of >10%, voriconazole-echinocandin combination therapy or liposomal amphotericin B should be used as the initial therapy [52]. However, in many countries, there are no surveillance systems in place to determine the prevalence of triazole resistance in *A. fumigatus*, which is known to be the most common *Aspergillus* spp. causing IA, as has also been observed in the CAPA cases reported to date.

Rutsaert et al. from the Netherlands reported administering prophylactic aerosolised liposomal amphotericin-B to all COVID-19 patients on mechanical ventilation in critical care, after they identified a cluster of seven CAPA cases, four of which were proven. Antifungal prophylaxis formed part of a multi-faceted management of this cluster, which also included the bi-weekly GM screening of serum and BALF, if and when a bronchoscopy was performed. High-efficiency particulate air filters were also installed in their critical care unit. The authors reported that no further cases were detected after the implementation of these measures at the time of writing. The rationale for prospective trials would need to be determined in order to establish whether antifungal prophylaxis in severe COVID-19 cases is indicated. A clinical trial of posaconazole prophylaxis for the prevention of pulmonary aspergillosis in patients with severe influenza (NCT03378479) is currently ongoing and this will provide data on the effectiveness of this approach, at least for influenza [64].

New antifungal agents with novel modes of action are in the pipeline so as to address the problem of antifungal resistance, which threatens the effectiveness of the few agents currently being used to treat invasive fungal disease [65]. Clinical trials are ongoing for three new antifungal agents, namely, ibrexafungerp (NCT03672292) [66], olorofim (NCT03583164) [67], and fosmanogepix (NCT04240886) [68]. Ibrexafungerp, which is structurally similar to echinocandins, inhibits fungal β-1,3-glucan synthase with activity against triazole-resistant *Aspergillus* sp. Olorofim and fosmanogepix have different novel targets, which are fungal dihydroorotate dehydrogenase, an important enzyme in fungal DNA synthesis, and the inhibition of fungal enzyme Gwt1 inactivating modification of mannoproteins, which is an important component in maintaining fungal cell wall integrity, respectively [69,70]. All three agents have activity against *Aspergillus* spp., including *A. fumigatus,* which may impact positively on the future management of patients with IA and more specifically CAPA.

## 5. Conclusions

This review has highlighted the diagnostic and therapeutic challenges of CAPA, a newly identified fungal co-infection in patients with severe COVID-19. We underline the pitfalls of the current definitions of IA applied to these patients, and the need for further evaluation of the usefulness of the culture and detection of fungal antigens from upper respiratory tract specimens in the diagnosis of IA. Additionally, given the global emergence of triazole resistance in *Aspergillus* spp., performing AFST by phenotypic methods and/or the detection of mutations associated with antifungal resistance by genotypic methods is crucial to allow for the timely institution of appropriate antifungal therapy, and will provide valuable information on the prevalence of triazole resistance in *A. fumigatus* and other *Aspergillus* spp. for surveillance purposes. Furthermore, properly designed trials are needed in order to determine the optimum therapeutic approach for patients with CAPA.

## Figures and Tables

**Table 1 jof-06-00115-t001:** Categorization of the 38 published coronavirus disease 2019 (COVID-19) associated invasive pulmonary aspergillosis (CAPA) cases utilizing published definitions for invasive aspergillosis, and their treatment and outcome.

Author/ Country (Prevalence) [Ref]	Age/ Sex	Underlying Conditions	Local/Systemic CS Use	GM (ODI)/Serum BDG (pg/mL)/qPCR	Species (Triazole Susceptibility Pattern)	Expert Panel Case Definition of CAPA [35]	Bulpa et al. [36]	EORTC/MSGERC [37]	*Asp*ICU [38]	Treatment	Outcome
Koehler et al. Single center, retrospective Germany (5/19; 26.3%) [11]	62/F	Cholecystectomy for cholecystitis, arterial hypertension, obesity with sleep apnea, hypercholesterolemia, ex-smoker, COPD (GOLD 2)	Inhaled steroids for COPD	GM Serum negative/GM BALF> 2.5/qPCR BALF = positive	*A. fumigatus* (S) culture from BALF	Probable	Probable	N/A	Putative	VCZ	Died
	70/M	Vertebral disc prolapse left L4/5, ex-smoker	No	GM Serum = 0.7/GM BALF > 2.5/qPCR BALF = positive	Negative culture	Probable	N/A	N/A	N/A	ISA	Died
	54/M	Arterial hypertension, diabetes mellitus, aneurysm coiling	IV CS therapy 0.4 mg/kg/d, total of 13 days)	GM Serum negative/GM BALF > 2.5/qPCR BALF = positive	*A. fumigatus* (S) culture from ETA, ICZ 0.380 µg/mL, VCZ 0.094 µg/mL	Probable	N/A	N/A	Colonisation	CASPO → VCZ	Alive
	73/M	Arterial hypertension, bullous emphysema, smoker, COPD (GOLD 3), previous hepatitis B	Inhaled steroids for COPD	GM Serum negative/qPCR ETA = positive	*A. fumigatus* (S) culture from ETA, ICZ 0.380 µg/mL, VCZ 0.094 µg/mL	N/C	N/C	N/A	Colonisation	VCZ	Died
	54/F	None	No	GM Serum = 1.3 and 2.7qPCR ETA = negative	Negative culture	Probable	N/A	N/A	N/A	CASPO → VCZ	Alive
Alanio et al. Single center prospective France (9/27; 33.3%) [12]	53/M	Hypertension, obesity, ischemic heart disease	Dexamethasone IV 20 mg once daily from day 1 to 5 followed by 10 mg once daily from day 6 to 10	GM Serum = 0.13/GM BALF = 0.89/BDG = 523/qPCR BALF and serum = negative	Negative culture	N/C	N/A	N/A	N/A	None	Alive
	59/F	Hypertension, obesity, diabetes	No	GM Serum = 0.04/GM BALF = 0.03/qPCR BALF = negative	*A. fumigatus*, culture from BALF	Probable	N/A	N/A	Putative	None	Alive
	69/F	Hypertension, obesity	Dexamethasone IV 20 mg once daily from day 1 to 5, followed by 10 mg once daily from day 6 to 10	GM Serum = 0.03/BDG = 7.8/qPCR ETA = 23.9/qPCR serum negative	*A. fumigatus*, culture from ETA	N/C	N/A	N/A	Colonisation	None	Alive
	63/F	Hypertension, diabetes, ischemic heart disease	Dexamethasone IV 20 mg once daily from day 1 to 5, followed by 10 mg once daily from day 6 to 10	GM Serum = 0.51/GM BALF= 0.15/BDG = 105/qPCR BALF and serum = negative	Negative culture	Probable	N/A	N/A	N/A	None	Died
	43/M	Asthma with steroid use history	No	GM Serum = 0.04/GM BALF = 0.12/BDG = 7/qPCR BALF and serum = negative	*A. fumigatus*, culture from BALF	Probable	N/A	N/A	Putative	None	Alive
	79/M	Hypertension	Dexamethasone IV 20 mg once daily from day 1 to 5, followed by 10 mg once daily from day 6 to 10	GM Serum = 0.02/GM BALF = 0.05/BDG = 23/qPCR BALF = 34.5/qPCR serum = negative	*A. fumigatus*, culture from BALF	Probable	N/A	N/A	Putative	None	Alive
	77/M	Hypertension, asthma	Dexamethasone iv 20 mg once daily from day 1 to 5, followed by 10 mg once daily from day 6 to 10	GM Serum = 0.37/GM BALF = 3.91/BDG = 135/qPCR BALF = 29/qPCR serum = negative	*A. fumigatus*, culture from BALF	Probable	N/A	N/A	Putative	VCZ	Died
	75/F	Hypertension, diabetes	Dexamethasone iv 20 mg once daily from day 1 to day 5, followed by 10 mg once daily from day 6 to day 10	GM Serum = 0.37 GM BALF = 0.36 BDG = 450 qPCR BALF = 31.7 qPCR serum = Negative	*A. fumigatus*, culture from BALF	Probable	N/A	N/A	Putative	CASPO	Died
	47/M	Multiple myeloma with steroid therapy	No	GM Serum = 0.09 BDG = 14 qPCR ETA and serum = Negative	*A. fumigatus*, culture from ETA	N/C	N/A	Probable	Colonisation	None	Died
Van Arkel et al. Single center prospective Netherlands (6/31; 19.4%) [17]	83/M	Cardiomyopathy	Prednisolon 0.13 mg/kg/day for 28 days pre-admission	GM Serum = 0.4	*A. fumigatus*, culture from ETA	N/C	N/A	N/A	Colonisation	VCZ + ANID (5/6) L-AmB (1/6)	Died
	67/M	COPD (GOLD 3), Post RTx NSCLC 2014	Prednisolon 0.37 mg/kg/day for 2 days pre-admission	Not reported	*A. fumigatus*, culture from ETA	N/C	Possible	N/A	Colonisation		Died
	75/M	COPD (GOLD 2a)	No	GM BALF = 4.0	*A. fumigatus*, culture from BALF	Probable	Probable	N/A	Putative		Died
	43/M	None	No	GM Serum = 0.1 GM BALF = 3.8	Negative culture	Probable	N/A	N/A	N/A		Alive
	57/M	Bronchial asthma	Fluticason 1.94 mcg/kg/day for 1 month pre-admission	GM Serum = 0.1 GM BALF = 1.6	*A. fumigatus*. culture from BALF	Probable	N/A	N/A	Putative		Died
	58/M	None	No	Not reported	*Aspergillus spp.* culture from sputum	N/C	N/A	N/A	Colonisation		Alive
Rutsaert et al. Single center prospective Belgium (7/20; 35%) [13]	86/M	Hypercholesterinemia	No	GM serum = 0.1	*A. flavus* culture from ETA	N/C	N/A	N/A	Colonisation	None	Died
	38/M	Obesity, hypercholesterinemia	No	GM serum = 0.3 GM BALF > 2.8	*A. fumigatus* culture from BALF	Proven	N/A	Proven	Proven	VCZ, ISA	Alive
	62/M	Diabetes	No	GM serum = 0.2 GM BALF = 2	*A. fumigatus* culture from BALF	Proven	N/A	Proven	Proven	VCZ	Died
	73/M	Diabetes, obesity, hypertension, hypercholesterinemia	No	GM serum = 0.1 GM BALF > 2.8	*A. fumigatus* culture from BALF	Proven	N/A	Proven	Proven	VCZ	Alive
	77/M	Diabetes, chronic kidney disease, hypertension, pemphigus foliaceus	No	GM serum = 0.1 GM BALF = 2.79	*A. fumigatus* culture from BALF	Proven	N/A	Proven	Proven	VCZ	Alive
	55/M	HIV, hypertension, hypercholesterinemia	No	GM serum = 0.80 GM BALF = 0.69	Negative culture	Probable	N/A	N/A	N/A	VCZ, ISA	Died
	75/M	Acute myeloid leukemia	No	GM BALF = 2.63	*A. fumigatus* culture from BALF	Probable	N/A	N/A	Putative	VCZ	Died
Blaize et al. Case Report France (1) [19]	74/M	Myelodysplastic syndrome, CD8^+^ T-cell lymphocytosis, Hashimoto’s thyroiditis, hypertension, benign prostatic hypertrophy	No	Serum GM, BDG and qPCR negative, GM First ETA = negative First qPCR ETA = positive Second qPCR ETA = positiveDirect smear of the second ETA = branched septate hyphae	*A. fumigatus*, culture of second ETA	N/C	N/A	N/A	Colonisation	None	Died
Lescure et al. Case Series France (1/5; 20%) [14]	80/M	Thyroid cancer 2010 (patient presented with ARDS)	No	Not reported	*A. flavus*, culture from ETA	N/C	N/A	N/A	Colonisation	VCZ → ISA	Died
Antinori et al. Case Report Italy (1) [18]	73/M	Diabetes, hypertension, obesity, hyperthyroidism, atrial fibrillation	No	GM Serum = 8.6 qPCR from paraffin block tissue = positive	*A. fumigatus*, culture from BALF	Proven	N/A	Proven	Proven	L-AmB → ISA	Died
Prattes et al. Case Report Austria (1) [20]	70/M	COPD (GOLD 2), obstructive sleep apnea syndrome, insulin-dependent type 2 diabetes with end organ damage, arterial hypertension, coronary heart disease, obesity	Inhaled Budesonide (400 mg per day)	GM Serum = negative BDG = negative LFD positive from ETA	*A. fumigatus*, (S) culture from ETA VCZ = 0.125 µg/mL	N/C	N/A	N/A	Colonisation	VCZ	Died
Lahmer et al. Case Series Germany (2) [22]	80/M	Suspected pulmonary fibrosis	No	GM Serum = 1.5 GM BALF = 6.3	*A. fumigatus*, culture from BALF	Probable	N/A	N/A	Putative	L-AmB	Died
	70/M	None	No	GM Serum = negative GM BALF = 6.1	*A. fumigatus*, culture from BALF	Probable	N/A	N/A	Putative	L-AmB	Died
Meijer et al. Case Report Netherlands (1) [21]	74/F	Polyarthrosis, reflux, stopped smoking 20 years ago	No	GM serum = persistently < 0.5 GM ETA ≥ 3 BDG = 1590	*A. fumigatus*, culture from ETA (R)^TR34/L98H^ICZ = 16 µg/mL, VCZ = 2 µg/mL, and POSA = 0.5 µg/mL	N/C	N/A	N/A	Colonisation	VCZ + CASPO → Oral VCZ → L-AmB	Died
Mohamed et al. Case Report Ireland (1) [23]	66/M	Obesity, diabetes mellitus, hypertension, stopped smoking >10 years ago	No	GM serum = 1.1 GM ETA = 5.5 BDG = 202 qPCR ETA–*A. fumigati* complex	*A. fumigatus* culture from ETA (R) ^TR34/L98H^ICZ ≥ 32 µg/mL, VCZ = 2 µg/mL and POSA = 1 µg/mL	Probable	N/A	N/A	Colonisation	L-AmB	Died
Sharma et al. Case Report Australia (1) [24]	66/F	Hypertension, recent ex-smoker of 20 pack years	No	Not done	*A. fumigatus* from ETA	N/C	N/A	N/A	Colonisation	VCZ	Alive
Santana et al. Case Report Brazil (1) [15]	71/M	Hypertension, diabetes mellitus, chronic kidney disease	No	GM stored blood 4.29 qPCR of lung tissue, Sequencing identified *Aspergillus penicillioides*	Not done	Proven	N/A	Proven	Proven	None	Died
Ferreira et al. Case Report France (1) [16]	56/M	Hypertension, diabetes mellitus, hyperlipidemia, obesity	Fluticasone propionate/ salmeterol inhaler, Dexamethasone IV 20 mg *×* 7 days	GM serum First sample = 0.07, Second sample = 0.05 BDG First sample = 10.4, Second sample ≤ 7.8 qPCR ETA 26.3 qPCR serum negative	*A. fumigatus*, culture from ETA (R)^TR34/L98H^ICZ = >8 µg/mL, VCZ = 2 µg/mL, ISA 4 ug/mL and POSA = 0.5 µg/mL	N/C	N/A	N/A	Colonisation	None	Died

Legend: N/A, not applicable; N/C, not classifiable; M, male; F, female; IV intravenous; CS corticosteroids; BALF bronchoalveolar lavage fluid; ETA, endotracheal aspirate; GM, galactomannan; ODI, optical density index; qPCR, quantitative polymerase chain reaction; BDG, 1-3 β-d-glucan; LFD, *Aspergillus* lateral flow device; ICZ itraconazole; VCZ, voriconazole; ISA, isavuconazole; POSA, posaconazole; CASPO, caspofungin; L-AmB, liposomal amphotericin-B; S, susceptible; r, resistant; COPD, chronic obstructive pulmonary disease; GOLD, Global Initiative for Chronic Obstructive Lung Disease; RTx, radiotherapy; NSCLC, non-small cell lung cancer; ARDS, adult respiratory distress syndrome.

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
