# Peer review of "COVID-19 Associated Invasive Pulmonary Aspergillosis: Diagnostic and Therapeutic Challenges"

_jof, 2020, doi:10.3390/jof6030115_

Round 1
Reviewer 1 Report
The authors of this review decribe the difficulties in the diagnosis of invasive aspergillosis in patients with severe COVID-19. The manuscript is well writen, with a clear introduction and well structure main text. The authors make appropiate use of tables, which are readable. Here are some suggestions to further improve the manuscript.
- only few cases have been published to date. incidence of CAPA vary in these reports. It is not clear what the true incidence of CAPA is. I would suggest a sentence/parts eloborating further on the incidence/lack of knowledge of the incidence of CAPA complicating COVID-10
- The proven cases show that some patients do have invasive aspergillosis. However, do all patients with growht of Aspergillus in or positive galactomannan in the respiratory tract have invasive disease or it is possible that some patients are colonized with aspergillus without invasive disease. In addition to the challenges to diagnose CAPA in patients with COVID-19, differentiating invasive disease and colonization is another challenge. However, some patients survive without antifungal treatment. I would suggest to further discuss the problems to differentiate between colonisation and infection. - Is positive serum GM a strong marker for invasive disease? - Can we exclude CAPA when serum GM is negative (not to this reviewers opinion). - is there already any data suggesting antifungal treatment is improving patient survival?
- The authors suggest to treat all patients with 2 mycological criteria. I suggest to give further argumentation for this statement and include the uncertainties of points 1 and 2. Do the authors think the strenght of the suggestion to treat the patients should be distinguised between serum GM positive and serum GM negative patients or is there not enough data at this point?
Author Response
Response to Reviewer 1
The authors of this review describe the difficulties in the diagnosis of invasive aspergillosis in patients with severe COVID-19. The manuscript is well wriiten, with a clear introduction and well structure main text. The authors make appropiate use of tables, which are readable.
Thank you.
Here are some suggestions to further improve the manuscript.
- Only few cases have been published to date. The incidence of CAPA vary in these reports. It is not clear what the true incidence of CAPA is. I would suggest a sentence/parts eloborating further on the incidence/lack of knowledge of the incidence of CAPA complicating COVID-10.
The text has been revised accordingly. Please see line 80 page 2.
- The proven cases show that some patients do have invasive aspergillosis. However, do all patients with growth of Aspergillus in or positive galactomannan in the respiratory tract have invasive disease or it is possible that some patients are colonized with aspergillus without invasive disease. In addition to the challenges to diagnose CAPA in patients with COVID-19, differentiating invasive disease and colonization is another challenge. However, some patients survive without antifungal treatment. I would suggest to further discuss the problems to differentiate between colonisation and infection. - Is positive serum GM a strong marker for invasive disease? - Can we exclude CAPA when serum GM is negative (not to this reviewers opinion). - is there already any data suggesting antifungal treatment is improving patient survival?
We agree with all the points raised by the reviewer. We currently do not know the true incidence of CAPA, which maybe due to lack of awareness, lack of screening and because of the difficulty of diagnosis and differentiating infection to that of colonisation. Similarly we agree that a negative serum GM does not outrule invasive pulmonary aspergillosis as seen in non-neutropenic patients who more often have negative serum GM. We do not know as yet as well whether treatment improves survival which we have discussed in the manuscript. The manuscript was revised and a reference (number 41) added. Please see line 35 – 37 page 11.
- The authors suggest to treat all patients with 2 mycological criteria. I suggest to give further argumentation for this statement and include the uncertainties of points 1 and 2. Do the authors think the strength of the suggestion to treat the patients should be distinguised between serum GM positive and serum GM negative patients or is there not enough data at this point?
Again, we agree with the reviewer, Due to the difficulty in confirming the diagnosis, we suggested that patients who meet at least 2 mycological criteria which can be a combination of culture, galactomannan (serum/BALF/ETA), PCR or BDG be treated with antifungal agents. At present, there is not enough data to show the benefit of this approach as we discussed in our manuscript.
Submission Date
Reviewer 2 Report
Abstract:
The abbreviation CAPA needs to be defined.
Introduction:
Well-balanced and very informative leading the reader to the point of the review.
Text sections:
Diagnosis of CAPA
The section refers to IA which conflicts with the title that refers to PA. However, PA is a umbrella term for chronic (CPA: CCPA, CFOPA,m SAIA, aspergillomas, aspergillus-related nodules) as well as invasive (IPA: invasive pulmonary aspergillosis) subtypes. In such, the title should be more specific to include “… associated invasive pulmonary aspergillosis…” In prolongation hereof, IPA should be consistently used in that context. As it appears now there is a switch between IA and IPA. Seldomly IA is not matching with IPA (a rare case could be a brain aspergilloma in an immunosuppressed patient without pulmonary pathology developing to IA). (Please see: Kosmidis C, Denning DW. Thorax 2015;70:270–277).
Reversed halo sign (= Atoll sign) is more a radiological indicator of organizing pneumonia which is seen in different types of pneumonia including CPA and IPA. However, a patognomonic radiological sign of angioinvasive aspergillosis (IA) is the halo sign (nodules or minor infiltrates surrounded by “clouds” of ground glass opacities). I prefer this to be specified with relevant references.
Page 12, line 71-75: Could be easier to read if use of bullets. Now it is a quite long sentence.
AFT strategies for CAPA
It should be emphasized that reference 50-52 concerns present recommendations on “solely” IPA. Though it is issued later in this text section, the reader may be confused whether these references regard to CAPA. Moreover, it should also be emphasized that the already challenging AFT regarding IPA is only going to be more complex, when it appears secondary to SARS-CoV-2 related infection/ARDS. This would support the following endorsement for further phenotype based AFT.
Along with the mentioning of other authors’ recommendation on future studies on CAPA AFT, the authors are also recommended to come up with such a statement, and also how future research could clarify this challenging issue (RCT with more arms?).
Other comments:
The Introduction very nicely describes how the immunological system may conduct lung damage when being SARS-CoV-2 infected. In this context, a minor section could advantageously be embedded on potential causes/reflections/speculations to why somer patients reacting by severe SARS-CoV-related pneumonia/hypoxeamia/ARDS may be more prone to IPA?
Author Response
Reviewer Number 2
We thank Reviewer 2 for reviewing our manuscript. We note the recommended changes which we have addressed accordingly. Our reply is in italics.
Abstract:
The abbreviation CAPA needs to be defined.
This has been changed accordingly in the abstract.
Introduction:
Well-balanced and very informative leading the reader to the point of the review.
Thank you.
Text sections:
Diagnosis of CAPA
The section refers to IA which conflicts with the title that refers to PA. However, PA is a umbrella term for chronic (CPA: CCPA, CFOPA,m SAIA, aspergillomas, aspergillus-related nodules) as well as invasive (IPA: invasive pulmonary aspergillosis) subtypes. In such, the title should be more specific to include “… associated invasive pulmonary aspergillosis…” In prolongation hereof, IPA should be consistently used in that context. As it appears now there is a switch between IA and IPA. Seldomly IA is not matching with IPA (a rare case could be a brain aspergilloma in an immunosuppressed patient without pulmonary pathology developing to IA). (Please see: Kosmidis C, Denning DW. Thorax 2015;70:270–277).
We appreciate the comments and the title was changed as suggested.
Reversed halo sign (= Atoll sign) is more a radiological indicator of organizing pneumonia which is seen in different types of pneumonia including CPA and IPA. However, a patognomonic radiological sign of angioinvasive aspergillosis (IA) is the halo sign (nodules or minor infiltrates surrounded by “clouds” of ground glass opacities). I prefer this to be specified with relevant references.
The manuscript has been revised accordingly. Please see lines 58-60 page 12.
Page 12, line 71-75: Could be easier to read if use of bullets. Now it is a quite long sentence.
The manuscript has been revised as suggested by the reviewer. Please see lines 76 – 81 page 12
AFT strategies for CAPA
It should be emphasized that reference 50-52 concerns present recommendations on “solely” IPA. Though it is issued later in this text section, the reader may be confused whether these references regard to CAPA. Moreover, it should also be emphasized that the already challenging AFT regarding IPA is only going to be more complex, when it appears secondary to SARS-CoV-2 related infection/ARDS. This would support the following endorsement for further phenotype based AFT.
Along with the mentioning of other authors’ recommendation on future studies on CAPA AFT, the authors are also recommended to come up with such a statement, and also how future research could clarify this challenging issue (RCT with more arms?).
Thanks for these comments. References 50 – 52 are international guidelines on the treatment of invasive aspergillosis since this is a novel co-infection there are no guidelines for the treatment of CAPA. The concluding paragraph has been revised as suggested. Please see lines 149 – 150 page 13.
Other comments:
The Introduction very nicely describes how the immunological system may conduct lung damage when being SARS-CoV-2 infected. In this context, a minor section could advantageously be embedded on potential causes/reflections/speculations to why somer patients reacting by severe SARS-CoV-related pneumonia/hypoxeamia/ARDS may be more prone to IPA?
We note the reviewers’ comment however another article in the same issue has written a review focused on the immunology of CAPA. This is the reason why we focused on the diagnostic and therapeutic challenges in our review.